# Evaluation of Atmospheric Environmental Regulations: The Case of Thermoelectric Power Plants

**Mateo Concha [1] and Gonzalo A. Ruz [1,2,3,\*]**

[1]   Facultad de Ingeniería y Ciencias, Universidad Adolfo Ibáñez, Diagonal Las Torres 2640, Peñalolén, Santiago 7941169, Chile
[2]   Center of Applied Ecology and Sustainability (CAPES), Santiago 8331150, Chile
[3]   Data Observatory Foundation, Santiago 7941169, Chile
[\*]   Correspondence: gonzalo.ruz@uai.cl

**Abstract:** In Chile, the concept of sacrifice zones corresponds to those land surfaces in which industrial development was prioritized over the environmental impact that it caused. A high number of industries that emit pollutants into the environment are concentrated in these zones. This paper studies the atmospheric component of the Environmental Impact Declaration and Assessment's (EID and EIA, respectively) environmental assessment instruments of the thermoelectric power plants in northern Chile, based on their consistency with current environmental quality regulations. We specify concepts on air quality, atmospheric emission regulations, and the critical parameters and factors to be considered when carrying out an environmental impact assessment. Finally, we end by presenting possible alternatives to replace the current methodologies and criteria for atmospheric regulation in areas identified as saturated or of environmental sacrifice, with an emphasis on both population health and an environmental approach.

**Keywords:** power plant; atmosphere; pollution; regulation; policy; coal-fired; emission





## 1. Introduction

### 1.1. Impact and Environmental Regulation

Power generation corresponds to one of the largest industries in Chile, playing an important role in the development of the economy, contributing 7.2% to the national GDP in 2018 [1]. Given the different production and generation processes carried out, a wide variety of waste is generated in the different productive sectors, classified according to the receiving environment they affect (soil, water, or air). Out of the consequences of the industrial processes of energy generation, the most important one is the atmospheric emission of exhaust gases resulting from the burning of fossil fuels [2], since these are dispersed at the mercy of the wind once they leave the chimney, carrying pollutants such as sulfur dioxide, nitrogen dioxide and respirable particulate matter.

Continued exposure to these conditions is harmful to both humans and the environment. For humans, it can be fatal in the long term because it affects the cardiac, respiratory, neurological, and endocrine systems [3]. According to a study developed by the Department of Epidemiology and Public Health of the University of Basel, Switzerland, diabetes seems to be induced after long-term air pollution exposure. Continuous exposure to polluted air has been directly linked to some types of cancer, such as lung or throat cancer [4]. For the environment, the most important effects of continuous exposure to polluted air are acid rain, greenhouse effect, depletion of ozone, and in specific scenarios it can cause reproductive failure in wildlife [5].

Furthermore, multiple studies have demonstrated that constant exposure to environments with high loads of SO and/or NO affects different materials through corrosion of their surfaces [6–9]. Saleh IA (2021) demonstrated that a higher level of corrosion in building materials was found for samples exposed to polluted air in industrial and urban

areas compared to those that were exposed to rural zones, linking the first ones to higher concentrations of pollutants and particles in suspension [10].

Given the different damages that this phenomenon causes, different global entities, such as the Organization for Economic Co-operation and Development (OECD) and the World Health Organization (WHO), have implemented different standards that seek to regulate the atmospheric emissions emitted by the energy industry to protect the health of people and the environment close to the emission sources [11].

At the national level, in the 1990s, the first law in Chile that ensured the regulation of the environment came into effect: Law 19.300, on General Bases of the Environment, which establishes the general framework of regulation, protection, and preservation of nature and the environment, as well as dictates the control guidelines of the Environmental Impact Assessment System [12]. This law contemplated the fundamentals of a normative process for environmental regulation in Chile, forcing the actors to understand and comprehend the terms and concepts in the way that the legislator determined them. Among them, the environment, biodiversity, pollution, conservation, preservation, impact, and environmental damage stood out, as well as decreeing different environmental management instruments, that is, state tools for the administration and regulation of environmental components [13].

In 2010, the administrative mismanagement of the law [13,14] and the urgency of regularizing environmental legislation led to the creation of the Ministry of the Environment, the Environmental Assessment Service (EAS), and the Superintendence of the Environment (SE) as impartial monitoring agents of environmental regulations in Chile [15,16]. With the goal of protecting people's health and the environment, the Ministry of Environment reviewed the existing regulations and generated a new environmental normative to the emissions of each type of atmospheric pollutant from mobile sources (Varotsos C et al. (2012) studied the correlation between the high levels of NO and Ultra-Fine particles in air with near-constant traffic from city streets [17]) and stationary sources (manufacturing and electricity industries, construction activities, waste burning, combustion of oil, coal, and biomass in the households [18], establishing the maximum limits for each type of atmospheric pollutant emitted as a result of its operations that put the integrity of the environment at risk [19,20]).

In October 2011, Supreme Decree No. 40 was promulgated, which "Approves Regulations for the Environmental Impact Assessment System (EIAS)" [21] establishing the guidelines by which the EIAS will be governed and the participation of the community in the environmental impact assessment process, in accordance with the provisions of Law No. 19.300 [12]. It decrees the definitions that, in accordance with the General Bases of the Environment, will be used for the correct execution of the environmental assessment instruments.

- Area of influence: The geographic area or space, whose attributes, namely natural or sociocultural elements, must be considered to define if the project or activity generates or presents any of the effects, characteristics, or circumstances of article 11 of Law 19.300, or to justify the non-existence of effects, characteristics, or circumstances;
- Emission: Release or transmission to the environment of any pollutant by a project or activity;
- Environmental impact: Alteration of the environment, caused directly or indirectly by a project or activity in an area;
- Baseline: Detailed description of the elements within the area of influence in which the effects or impacts of project implementation will be generated.

Given the importance of these aspects, and with the aim of unifying criteria, procedures and methodologies to describe, characterize, and analyze the area of influence, in 2015, the Environmental Assessment Service published the "Guide for the description of air quality in the area of influence of the projects that enter the EIAS" [22], which presents the main guidelines to consider regarding the description of air quality in the area of influence of a new project that is presented to the EIAS, with the objective of being able to correctly

evaluate the impact that the atmospheric emissions generated by the emitting sources may cause prior to their execution.

*1.2. Power Generation and Atmospheric Emissions*

In the energy industry, the emission sources of atmospheric pollutants that generate the greatest environmental impact are thermoelectric plants [19], since these are facilities destined to the generation of electrical energy from the energy released by the burning of fossil fuels, corresponding to 97% of energy industry's greenhouse gases (GHG) emissions generated in 2016, equivalent to 34.6 [Mt $CO_2$eq] [23]. Taking into consideration that the total $CO_2$ emissions in Chile in 2016 were 88.5 [Mt $CO_2$eq], the emissions and processes of thermal power plants acquire great importance and must be continuously monitored by an auditing entity to comply with the regulations [23].

A Thermoelectric Power Plant (TPP) is a facility designed to generate electricity from the energy released by fossil fuels. In it, four macro stages can be identified: (i) heat generation by burning coal or diesel and generation of exhaust gases; (ii) closed water circuit, where liquid water is heated, transforming into steam at a high temperature which moves a turbine; (iii) open water circuit, where low temperature water is extracted, in most cases, from the ocean and is used to cool the steam that comes out of the turbine, which is then returned to the ocean; and (iv) the transformation in the generator of the kinetic energy generated in the turbine to electrical energy [24]. In this way, the emission of pollutants in the exhaust gases by the plant will vary in composition and quantity depending on the type of raw material combusted for power generation.

Finally, the creation in 2011 of Supreme Decree No. 13 [19], under the title "Establish Emission Standard for Thermoelectric Power Plants" marks a milestone in the energy generation industry from fossil fuels. Its main objective was to control the atmospheric emissions of Sulfur Dioxide ($SO_2$), Nitrogen Oxides (NOx), and Respirable Particulate Matter (PM10), the main pollutants from the operation of thermoelectric power plants [20], restricting the emission limits for sources existing and establishing new and more demanding limits for new thermal power plants, considering as an existing unit the one whose construction date was declared prior to 30 November 2010, with constructions from that date onwards considered as new emitting sources. However, the division by years of construction of the scope of this decree does not consider the fact that the past behavior of atmospheric pollutants has a strong correlation with the present state of the system [25].

This work analyzes the EIS and EIA of the thermoelectric power plants in northern Chile according to compliance with current environmental regulations. Then, the current environmental regulations in Chile will be studied, the thermal complexes will be identified as emission sources, and the pollutant emissions receptor points will be determined. This analysis will allow non-expert users to realize that the atmospheric component on the EIS and EIA does not have the basis to be used as a reliable air pollution data.

The main contributions of this study are as follows:

(i) It presents for the first time the crossing of information between the quality of records documented in the EIS and EIA, and the representativeness of air quality monitoring stations.

(ii) It highlights the shortcomings and loopholes found in the current air quality regulations regarding the representativeness of atmospheric records.

(iii) It shows the importance of the location of air quality monitoring stations in the representativeness of their records.

(iv) It reveals Chile's lack of awareness regarding the emission of atmospheric pollutants and, therefore, their consequences for health.

## 2. Materials and Methods

This study has a three-part methodology, which includes the review, evaluation, and application of the norms and regulations in force in Chile. The guideline of what was decreed in Art.10, letter c, of Law 19.300 was followed, which establishes that "Projects

and/or activities related to power generation plants greater than 3MW must be submitted to the environmental impact assessment system". A review of the Chilean government web-sites related to energy and the platforms of the EAS, the SE, and the National Information System for Environmental Enforcement (NISEE) was carried out.

For the identification of the energy-generating projects, all the documents available on the EAS digital platform uploaded with projects executed between the years 1990 and 2020, indexed in the "energy" productive sector and categorized in accordance with Decree Supreme 95 as "Power Generating Plants Greater than 3 MW", were obtained. Then, only projects where energy generation was derived from the burning of fossil fuels were selected/admitted for the current study. Out of the prior selection of projects, the projects whose status was "approved" by the inspection entity were chosen, and those thermal power plants belonging to the thermoelectric complexes with the highest installed capacity at the end of 2020 were selected [26].

Project information on the EAS platform can be found in two ways, as an Environmental Impact Declaration (EID) or as an Environmental Impact Assessment (EIA). Each environmental management instrument (EMI) submitted to the EIAS will correspond to a new project or a modification of a current one. In accordance with Art. 5 of the Regulation of the Environmental Impact and Assessment System [21], those responsible for a project that will be carried out in a territory must present an EIA if it generates or presents a risk to the health of the local population, because of the quantity and quality of the effluents, emissions, or waste that it generates or produces. On the contrary, all those in which its execution is not harmful to health must present an EID.

Depending on the type of instrument available in the file of each project and maintaining the main focus of the study regarding the regulation of emissions and air quality, the work was carried out as follows: For the EIDs, the background information presented to fulfill the environmental regulations in the area where the project would be executed was evaluated. In the case of the EIAs, which are more complex studies that provide the background for the prediction, identification, and interpretation of the environmental impact of the project activities, the background information presented to comply with the environmental quality regulations and those projections regarding the emissions that the operations of the projects would generate were evaluated.

For the classification of projects, it is necessary to introduce the concept of the "sacrifice zone". This is defined as a place that concentrates a large number of polluting industries, in which economic production is prioritized over the environmental impact that the operations of these will cause [27]. Since the emergence of this concept in the mid-1980s, several locations on the north of Chile have been identified with this concept: the bays of Mejillones, Tocopilla, Huasco, and Puchuncavi standing out for their criticality [28].

For the development of this study, the thermoelectric projects located in these four locations are chosen since they have the highest impact at the national level in the generation of atmospheric pollutants because of the production of electrical energy resulting from the burning of fossil fuels [29]. Finally, it is necessary to mention that all the information used in this study was not modified in any way. All the data, tables, statistics, and models are presented as they were presented to the environmental inspection entity for approval, considering them as real and reliable information.

## 3. Results

### 3.1. Environmental Regulations

In Chile, the SMA is the organism in charge of executing, organizing, and coordinating the monitoring and inspection of the content of environmental quality standards and gas emission standards [30]. As of June 2022, there are six emission standards which establish the maximum amount of a pollutant allowed, measured directly at the emission source, and nine air quality standards, presented in two categories: primary and secondary [12].

The SE presents six emission standards related to atmospheric components such as air, noise, and light pollution, establishing them as the general guidelines applicable to all industrial establishments in operation.

- D.S. No. 38/2011, Noise Standard;
- D.S. No. 43/2013, Light Pollution Standard;
- D.S. No. 28/2013, Foundry Standard;
- D.S. No. 13/2011, Standard for Thermoelectric Power Plants;
- D.S. No. 37/2013, TRS Compound Standard;
- D.S. No. 29/2013, Establishes Emission Standard for incineration, co-incineration and co-processing and repeals Decree No. 45, of 2007, of the Ministry General Secretariat of the Presidency.

These normatives aim to regulate the generation of emissions observed directly from the emission sources through the development and presentation of structured and homologous reports between emission sources to the inspection authorities, in which the project owners provide the information required to demonstrate compliance with emission limits for pollutants emissions from industrial operations. These regulations alone are not enough to keep the atmospheric component regulated, so they are complemented by primary and secondary quality standards.

The primary quality standards establish the values of the concentrations and permissible periods of the elements, compounds, substances, and chemical or biological derivatives, whose presence or lack in the environment may constitute a risk to the health of the population. The secondary standards of environmental quality establish the same parameters while ensuring the conservation and protection of the environment.

Below are the primary air quality standards, currently in July 2022:

- Air quality standard for PM2.5 [31];
- Air quality standard for PM10 [32];
- Air quality standard for $SO_2$ [33];
- Air quality standard for $NO_2$ [34];
- Air quality standard for CO [35];
- Air quality standard for Pb [36];
- Air quality standard for $O_3$ [37].

Next, for each regulated pollutant, the atmospheric concentration ranges that cause environmental emergency situations are presented and calculated as the moving average of the observed period (see Table 1).

**Table 1.** Contaminant concentration and environmental emergency levels.

| Pollutant | Period | Level | Concentration ($\mu g/m^3 N$) | Reference |
|---|---|---|---|---|
| PM 2.5 | 24 h | Alert Pre-emergency Emergency | 80–109 110–169 170 or + | [31] |
| PM10 | 24 h | Alert Pre-emergency Emergency | 195–239 240–329 330 or + | [32] |
| $SO_2$ | 1 h | Alert Pre-emergency Emergency | 500–649 650–949 950 or + | [33] |
| $NO_2$ | 1 h | Alert Pre-emergency Emergency | 1130–2259 2260–2999 3000 or + | [34] |
| CO | 8 h | Alert Pre-emergency Emergency | 17–33 34–39 40 or + | [35] |
| $O_3$ | 1 h | Alert Pre-emergency Emergency | 400–799 800–999 1000 or + | [37] |

In the case of atmospheric lead [36], the primary standard establishes that the limit will be 0.5 ($\mu$g/m$^3$N) as an annual concentration, considering this an environmental emergency when the arithmetic average of the concentration values of two successive years exceeds that concentration for the same monitoring station. Additionally, it is necessary to highlight that to date, there is no quality standard that regulates the concentrations of hydrocarbon compounds or their derivatives in Chile.

Additionally, secondary air quality standards, in July 2022 in Chile are as follows:

- Air quality standard for SO$_2$ [38];
- MPS air quality standard in Huasco [39].

For the purposes of applying the air quality standard for sulfur dioxide, the country is divided into northern and southern zones, establishing its geographical limits in Article 3 of Supreme Decree 22 of 2009 [38]. Table 2 presents the limit concentrations according to the measurement period for the two zones.

**Table 2.** Limit concentrations of sulfur dioxide.

| Pollutant | Period | Concentration Northern Zone ($\mu$g/m$^3$N) | Concentration Southern Zone ($\mu$g/m$^3$N) | Reference |
|---|---|---|---|---|
| SO$_2$ | Annual | 80 | 60 | Art 4 [38] |
| | 24 h | 365 | 260 | Art 5 [38] |
| | 1 h | 1000 | 700 | Art 6 [38] |

The second secondary norm, referring to the concentrations of sedimentable particulate matter (SPM), is limited to the Huasco River basin in the third region of Chile. It establishes the maximum permissible values, calculated as a monthly arithmetic mean for SPM as 150 mg/m$^2$ per day and 60 mg/m$^2$ per day for iron in SPM. Likewise, it establishes the limits calculated as an annual arithmetic mean at 100 mg/m$^2$ per day for SPM and at 30 mg/m$^2$ per day for iron in SPM.

### 3.2. Thermal Emitting Complexes

The information of the projects of the EAS platform was downloaded under the criteria of an approved project and belonging to the productive energy sector, delivering a total of 2019 projects found at the national level. Of the total, all those that met typology "c" were selected, those being power generation plants greater than 3 MW, established in DS95 [26] for high-voltage power transmission lines and their substations, reducing the number to 511 projects submitted to the EIAS. Finally, the power generation projects based on the burning of fossil fuels that were in the most critical sacrifice zones in the north of the capital were filtered. For the town of Tocopilla, nine projects were identified, fifteen for Mejillones, eleven for Huasco, while for the town of Puchuncavi there were a total of five projects, arriving at a total of forty projects and/or modification of one (see Table 3).

**Table 3.** List of projects and/or modifications.

| Year | Central | Project Name | Type | Commune | Latitude | Longitude |
|---|---|---|---|---|---|---|
| 1999 | Coloso | Central Termoeléctrica de Ciclo Combinado Coloso | EIA | Tocopilla | −23.77279 | −70.482385 |
| 1993 | Nueva Tocopilla | Central Termoeléctrica Nueva Tocopilla [1] | EIA | Tocopilla | −22.09617 | −70.210518 |
| 1993 | Nueva Tocopilla | Central Termoeléctrica Nueva Tocopilla 2 [1] | EIA | Tocopilla | −22.09573 | −70.209923 |
| 1996 | Nueva Tocopilla | Sistema de Transmisión Central Termoeléctrica Nueva Tocopilla [1] | EIA | Tocopilla | −22.09625 | −70.211992 |
| 1999 | Nueva Tocopilla | Aumento de potencia de generación y uso de mezclas de petcoke y carbón central termoeléctrica nueva Tocopilla [1] | EIA | Tocopilla | −22.09774 | −70.213217 |

**Table 3.** *Cont.*

| Year | Central | Project Name | Type | Commune | Latitude | Longitude |
|---|---|---|---|---|---|---|
| 1998 | Tocopilla | Central Termoeléctrica de Ciclo Combinado Tocopilla [1] | EIA | Tocopilla | −22.09779 | −70.212732 |
| 1999 | Tocopilla | Uso de Mezclas de Carbón y Coque de Petróleo como Combustible Central Tocopilla (2° presentación) [1] | EIA | Tocopilla | −22.09625 | −70.211992 |
| 2007 | Tocopilla | Operación permanente con petróleo diésel en la Unidad 16 [1] | EIS | Tocopilla | −22.09754 | −70.212488 |
| 2013 | Tocopilla | Uso de Cal Hidratada, Central Termoeléctrica Tocopilla para Cumplimiento de Norma de Emisión para Centrales Térmicas [1] | EIS | Tocopilla | −22.10077 | −70.213572 |
| 2006 | Andina | Central Térmica Andina[1] | EIA | Mejillones | −23.08942 | −70.412632 |
| 2007 | Andina | Embarcadero, Uso de Biomasa y Depósito de Cenizas Central Térmica Andino [1] | EIS | Mejillones | −23.12666 | −70.30776 |
| 2006 | Angamos | Central Termoeléctrica Angamos [1] | EIA | Mejillones | −23.07049 | −70.365255 |
| 2008 | Angamos | Modificación del Punto de Toma y Descarga Central Termoeléctrica Angamos [1] | EIS | Mejillones | −23.05782 | −70.373805 |
| 1997 | Atacama | Central Térmica Atacama [1] | EIA | Mejillones | −23.08787 | −70.414243 |
| 2002 | Atacama | Inclusión de By Pass de Gases y un Sistema de Aeroenfriadores en los Equipos Accesorios de una de las Turbinas a Gas de la Central Térmica Atacama Comuna de Mejillones II Región [1] | EIS | Mejillones | −23.09159 | −70.417532 |
| 2007 | Atacama | Obras Complementarias Central Atacama [1] | EIS | Mejillones | −23.08788 | −70.414613 |
| 2008 | Cochrane | Central Termoeléctrica Cochrane | EIA | Mejillones | −23.07314 | −70.36746 |
| 1996 | Mejillones | Central Termoeléctrica Mejillones Unidad 2 [1] | EIA | Mejillones | −23.08612 | −70.409343 |
| 1998 | Mejillones | Central Termoeléctrica Ciclo Combinado Mejillones CTM3 [1] | EIA | Mejillones | −23.08613 | −70.408367 |
| 2001 | Mejillones | Uso de un Combustible Alternativo en las Unidades 1 y 2 de la Central térmica Mejillones [1] | EIA | Mejillones | −23.09015 | −70.41291 |
| 2009 | Mejillones | Infraestructura Energética Mejillones [1] | EIA | Mejillones | −23.08640 | −70.405139 |
| 2014 | Mejillones | Uso de Cal Hidratada, Central Térmica Mejillones para Cumplimiento de Norma de Emisión para Centrales Termoeléctricas [1] | EIS | Mejillones | −23.08877 | −70.412309 |
| 2015 | Mejillones | Actualización Infraestructura Energética Mejillones [1] | EIS | Mejillones | −23.08640 | −70.405139 |
| 2013 | Ttanti | Central Termoeléctrica Ttanti | EIA | Mejillones | −23.09290 | −70.424595 |
| 1994 | Guacolda | Central Termoeléctrica Guacolda y Vertedero [1] | EIA | Huasco | −28.46511 | −71.255619 |
| 1999 | Guacolda | Usos de Mezclas de Carbón y Petcoke en Central Termoeléctrica Guacolda [1] | EIA | Huasco | −28.46511 | −71.255619 |
| 2004 | Guacolda | Flexibilización de la Operación en la Central Termoeléctrica Guacolda [1] | EIS | Huasco | −28.46397 | −71.257178 |
| 2005 | Guacolda | Central Guacolda Unidad N°3 [1] | EIA | Huasco | −28.46511 | −71.255619 |
| 2006 | Guacolda | Flexibilización Unidad N°3 [1] | EIS | Huasco | −28.46475 | −71.256245 |

**Table 3.** *Cont.*

| Year | Central | Project Name | Type | Commune | Latitude | Longitude |
|---|---|---|---|---|---|---|
| 2007 | Guacolda | Incremento de Generación y Control de Emisiones del Complejo Generador Central Térmica Guacolda [1] | EIA | Huasco | −28.46511 | −71.255619 |
| 2006 | Guacolda | Ampliación de la capacidad de almacenamiento de combustibles sólidos en la Central Térmica Guacolda [1] | EIS | Huasco | −28.46511 | −71.255619 |
| 2009 | Guacolda | Unidad 5 Central Térmica Guacolda S.A. [1] | EIA | Huasco | −28.46511 | −71.255619 |
| 2016 | Guacolda | Eliminación del uso de petcoke en central Guacolda y Ajuste de la Capacidad de Generación Eléctrica [1] | EIS | Huasco | −28.46496 | −71.256566 |
| 2009 | Maitencillo | Central Termoeléctrica Maitencillo | EIS | Huasco | −28.50853 | −70.91914 |
| 2012 | Punta Alcalde | Central Termoeléctrica Punta Alcalde | EIA | Huasco | −28.57843 | −71.2874 |
| 2012 | Campiche | Central Termoeléctrica Campiche [1] | EIA | Puchuncavi | −32.75278 | −71.478981 |
| 2005 | Nueva Ventanas | Central Termoeléctrica Nueva Ventanas (LFC) [1] | EIA | Puchuncavi | −32.7517 | −71.479378 |
| 2006 | Nueva Ventanas | Cambio de Combustible de la Central Termoeléctrica Nueva Ventanas [1] | EIS | Puchuncavi | −32.75029 | −71.483487 |
| 2007 | Nueva Ventanas | Ajuste de la disposición general de las instalaciones de la central nueva ventanas [1] | EIS | Puchuncavi | −32.75265 | −71.480044 |

[1] Only these approved projects were executed and are in operation.

The mentioned bays not only meet the condition of being sacrifice areas, but are also considered thermal generation complexes, that is, industrialized areas focused on the generation of electrical energy through coal-fire to the detriment of the quality of the environment [40]. For homologation in this study, from now on they will be referred to according to the corresponding sacrifice zone: Mejillones Thermal Complex (CTM), Tocopilla Thermal Complex (CTT), Huasco Thermal Complex (CTH), and Puchuncavi Thermal Complex (CTP).

*3.3. Station Registration*

Compliance with primary and secondary regulations is directly related to the representativeness of the monitoring stations, being representative for the population (EMRP) or for the natural resources (EMRRN) of the area where the air quality monitoring station is installed [41] since these deliver concentration data of gaseous pollutants registered in the location where they are set up. Presented in another way, the data from the monitoring station will provide sufficient information to identify whether or not an environmental quality standard is being met in that location, acquiring great relevance in the processes of evaluating regulations. Relevant data include Chile having 199 air quality monitoring stations, 71 state-owned and 128 privately owned [42], at the end of 2020.

The EIS and EIA studied in this investigation used the information registered in the air quality monitoring stations located according to Table 4 as support data for their evaluations.

The first thing that draws attention is the distribution of pollutants of interest by sacrifice zone. For CTM, the air quality monitoring stations are focused on recording the main atmospheric pollutants (PM10, PM2.5, $SO_2$ and NOx). In CTT the focus is concentrated on the presence of fine particulate matter and volatile sulfur dioxides. In Huasco Bay, the fact that nine of the eleven stations are capable of recording sulfur dioxide in the air stands out, five of them being unique for this function. Puchuncavi, located on the central coast of the country, has the largest number of pollutants under observation, with six monitoring stations, most of

which have the capacity to record all the pollutants generated by the industrial park in the area: PM10, PM2.5, $SO_2$, $NO_2$, NOx, NO, CO, $O_3$, $CH_4$, NMHC, and HTC.

**Table 4.** Monitoring stations and the pollutants that they are able to record.

| Commune | Station | E | N | PM10 | PM2.5 | SO$_2$ | NO$_2$ | NOx | NO | CO | O$_3$ | CH$_4$ | HCNM | HTC |
|---|---|---|---|---|---|---|---|---|---|---|---|---|---|---|
| Mejillones | Jardín Infantil Integra | 352,064 | 7,444,407 | x | | | x | x | x | | | | | |
| Mejillones | Juan Jose Latorre | 352,346 | 7,444,100 | x | x | x | x | x | x | | | | | |
| Mejillones | Compañía de Bomberos | 351,468 | 7,444,654 | x | | | x | x | x | x | x | | | |
| Mejillones | Ferrocarriles | 350,205 | 7,444,908 | x | x | x | x | x | x | x | x | | | |
| Tocopilla | Bomberos | 375,319 | 7,554,741 | | x | | | | | | | | | |
| Tocopilla | Gobernación | 376,284 | 7,556,725 | | x | | x | | | | | | | |
| Tocopilla | Super Site | 377,404 | 7,557,193 | x | x | x | x | x | x | x | x | | | |
| Tocopilla | Tres Marias | 377,485 | 7,560,029 | x | x | | | | | | | | | |
| Tocopilla | Escuela E10 | 377,352 | 7,557,219 | x | x | x | x | x | x | x | x | | | |
| Tocopilla | Centro | 376,516 | 7,556,334 | | | | x | | | | | | | |
| Tocopilla | Escuela E12 | 376,731 | 7,556,849 | x | | | x | | | | | | | |
| Tocopilla | Escuela Gabriela Mistral | 376,518 | 7,556,323 | | | | x | | | | | | | |
| Tocopilla | Sur | 374,794 | 7,554,836 | x | | | x | | | | | | | |
| Huasco | Huasco Sivica | 282,682 | 6,848,727 | | x | | | | | | | | | |
| Huasco | 21 de Mayo | 281,938 | 6,848,939 | x | x | | | | | | | | | |
| Huasco | EME F | 282,486 | 6,849,125 | x | x | x | x | | x | x | x | | | |
| Huasco | EME M | 282,763 | 6,848,691 | x | x | x | x | | x | | | | | |
| Huasco | EME ME | 279,008 | 6,849,199 | | | | x | | | | | | | |
| Huasco | Huasco II | 281,581 | 6,849,067 | | | | x | x | | x | | x | | |
| Huasco | SM1 | 279,357 | 6,845,277 | | | | x | | | | | | | |
| Huasco | SM2 | 286,412 | 6,849,717 | | | | x | | | | | | | |
| Huasco | SM3 | 286,750 | 6,848,592 | | | | x | | | | | | | |
| Huasco | SM4 | 287,841 | 6,847,562 | | | | x | | | | | | | |
| Huasco | SM5 | 289,916 | 6,847,254 | | | | x | | | | | | | |
| Puchuncavi | Puchuncaví | 274,379 | 6,377,331 | x | x | x | x | x | x | | x | | | |
| Puchuncavi | Campiche | 270,343 | 6,375,300 | x | | x | x | x | x | | x | x | x | x |
| Puchuncavi | La Greda | 268,185 | 6,373,910 | x | x | x | x | x | x | | x | | | |
| Puchuncavi | Los Maitenes | 270,073 | 6,372,171 | x | x | x | x | x | x | x | x | x | x | x |
| Puchuncavi | Ventanas | 267,547 | 6,374,609 | x | x | x | x | x | x | | x | x | x | x |
| Puchuncavi | Concentrados | 267,640 | 6,373,585 | x | | | x | x | x | | x | x | x | x |

Figure 1 indicates the Mejillones, Tocopilla, Huasco, and Puchuncavi bays in a map of Chile. The environmental impact statements and studies evaluated allowed the generation of zonal maps of the four thermal complexes, identifying the locations of both the thermoelectric plants and the environmental quality monitoring stations (Figures 2–5). The areas highlighted in green correspond to those in which the land use is considered residential use.

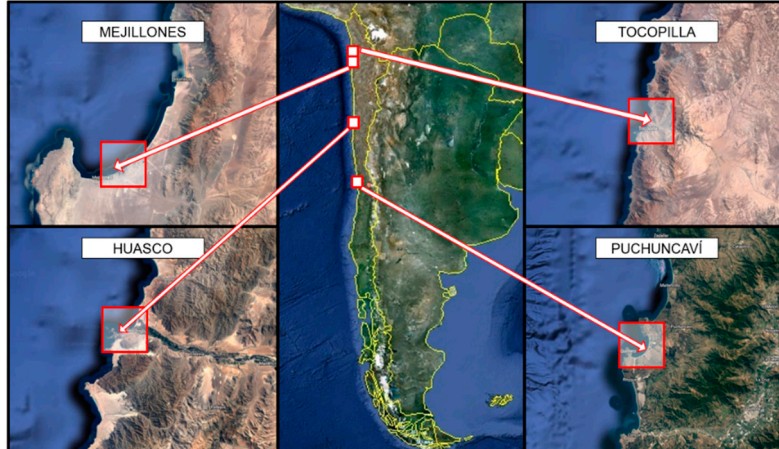

**Figure 1.** Location of Mejillones, Tocopilla, Huasco and Puchuncavi bays on a map of Chile.

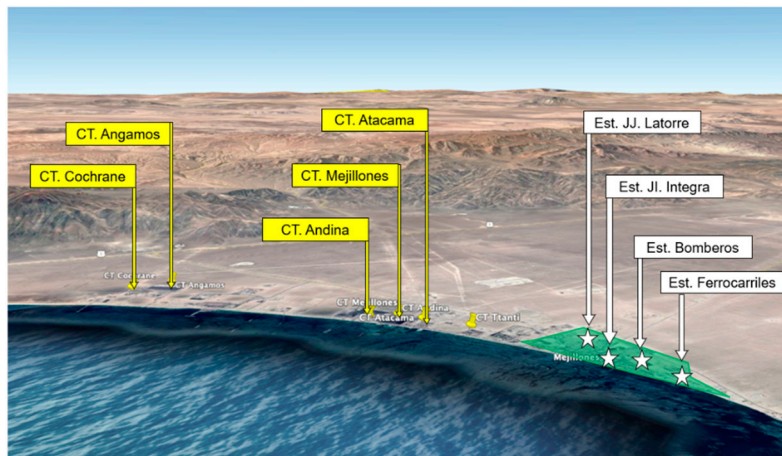

**Figure 2.** Distribution map of thermoelectric power plants, monitoring stations, and identification of urban settlements in Mejillones.

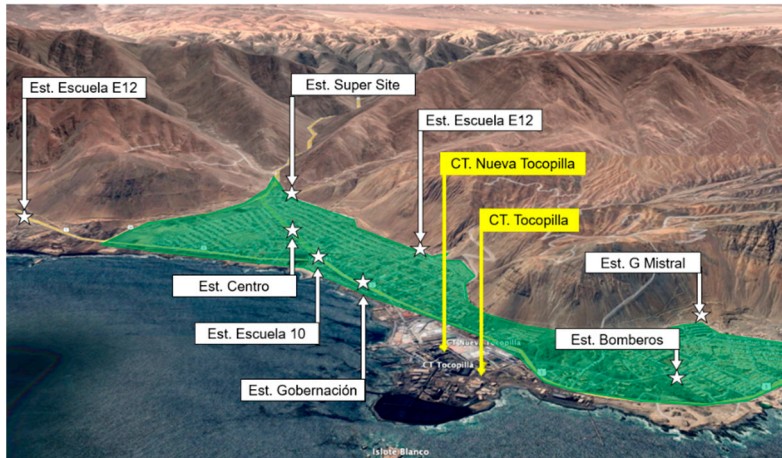

**Figure 3.** Distribution map of thermoelectric power plants, monitoring stations, and identification of urban settlements in Tocopilla.

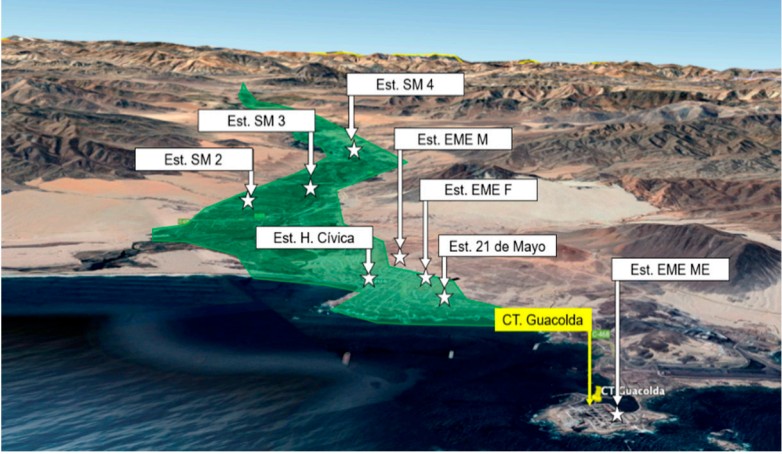

**Figure 4.** Distribution map of thermoelectric power plants, monitoring stations, and identification of urban settlements in Huasco.

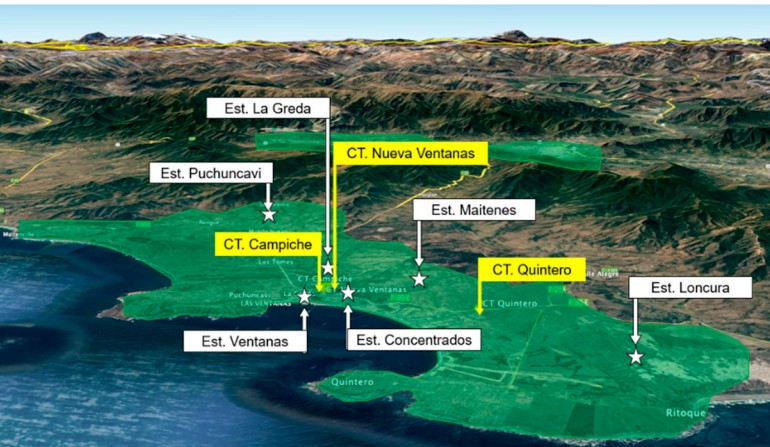

**Figure 5.** Distribution map of thermoelectric power plants, monitoring stations and identification of urban settlements in Puchuncavi.

## 4. Discussion

In 2022, the process of presenting information to the supervisory entities by the owners of power generation plants based on fossil fuels is fully complied with as decreed in the emission standard for thermoelectric plants [19], in which it is specified that the air quality baseline evaluations for the EISs, as well as the emission projections for the EIAs, the atmospheric data registered in the monitoring stations with active representativeness at the time of developing should be used as parameters studies [43].

EMRP and EMRRP monitoring stations are those that ensure that the information recorded represents a general condition of the area under observation, and that these are not the consequence of particular or isolated events, such as an industry that directly impacts the station, and does not impact the entire population of a location [22]. Under this definition, there are few stations presented in Table 3 that are capable of meeting this condition since, for the most part, they are located in the vicinity of large emission sources such as thermoelectric plants. On the other hand, even when this proximity condition may not be considered in the station-selection process, the information recorded in them must be carefully considered since, to comply with primary and secondary air quality regulations, the station must fulfill a double role of representativeness: from the population for primary standards and from natural resources for secondary standards [42]. Added to the restrictions of implementing an air quality station, the monitoring of pollutants continues to be a very technical and complex activity, so there are few operators specialized in providing the service of monitoring and retrieval of that information [22].

It shall be considered that most of the construction projects of a new power plant or generating unit were approved years before the concept of representativeness of monitoring stations [44] was established. Thus, its approval was conditional on the records of the air quality stations existing at that time, as well as whose information was available and accessible [45]. This included cases such as the one the EIA approved in 2006 regarding the Angamos Thermoelectric Power Plant, in which direct information from the air quality stations was not used to carry out the baseline survey or post-operation emission projections, but rather data were used that was indicated in other EIAs of industries in the area as a reference, considering that the information in them was accurate [46–48].

This type of practice is repeated in the EISs and EIAs evaluated, perhaps because of what was previously mentioned regarding the complexity of recovering the information from the existing monitoring stations. Therefore, the duplication of information together with the blind trust in the records captured by the stations leaves open the question of whether these data were correctly obtained, whether the reading of the station was continuous, or whether in the original study there was no tampering in their presentation. The problem can be translated into the fact that the information from the monitoring stations

available at the time of building an EMI is not sufficient to be used as a representative background of the atmospheric reality of the area where the project is to be implemented. Currently, the control of compliance with environmental regulations bases its validity on the quality of the environmental data recorded by the monitoring stations, based on what they deliver to the owners of the thermoelectric power plants to develop their operation and contaminant emissions plans.

As stated in the National Air Quality Information System (NAQIF), a station with representation will be one whose records represent the area where it is installed [44], so that the data collection location must be a strategic and calculated location to obtain the best possible record of the real situation of the area under observation. This statement borders on the obvious for optimizing results, but it was not until 2015 that the "Guide for the description of air quality in the area of influence of projects that enter the EIAS" [22] was published, in which the minimum considerations for the installation and use of the information registered in the monitoring stations were established for the first time, indicating that the location of the monitoring stations should be estimated in relation to the pollutant receptors that could potentially be affected for the operation of projects and/or industrial activities [21,49].

The lack of environmental priority given to these regulations has led to the approval of high atmospheric-impact projects for nearly three decades (the first Tocopilla thermal power plant was approved in 1994) based on data, without the certainty that these actually represent the environmental impact that their operations would cause.

Taking into account that (i) there is a problem with the current locations of the air quality stations, since these were not designated at the same time as the execution of the power generation projects, (ii) the information recorded in them is technical and highly complex, so there are few professional teams capable of interpreting it correctly, (iii) the updating of environmental air quality regulations is carried out based on the recorded data, (iv) the operation of thermoelectric power plants is supervised in accordance with the environmental quality regulations and atmospheric emissions, and (v) the approval of new projects and/or activities with an atmospheric impact is determined based on the current records captured by the monitoring stations, it is evident that if the data that are being used for decision-making is not coming from truly representative stations, the resulting decisions will not have real environmental bases to support their verdicts.

Carrying out the crossing exercise between the locations of the air quality monitoring stations and the thermoelectric projects in operation, there is a relationship based on compliance with the primary air quality regulations, which ensures the protection of the population health. However, when changing the criteria and observing from the point of view of compliance with secondary air quality regulations, it is not easy to make sense of the distribution of thermal power plants bases on environmental conservation.

The saturated zones presented in the results section show four bays inhabited and transformed into literal fixed sources of emissions on an environmental scale. Within the background evaluation carried out, it was possible to observe a great depth in the evaluation of the impacts of the operation of the plants on the populations and communities near the sites. However, the collection of information and critical analysis regarding the different aspects of the environmental impact was only sufficient to comply with the requirements of current regulations. With basic knowledge of the behavior of the wind, it is simple to identify that the clouds of pollutants emitted by the chimneys will have different behaviors according to the geography where the thermal power plant is installed. So, impact assessments should have a case-by-case approach to determine the real environmental impact that their emissions would cause.

One component that was not evaluated in any of the documents reviewed in this study was the impact of chimney emissions on the ocean. Although the impact of liquid effluents resulting from the open cycle of water in power plants has been extensively studied, including damage to marine flora and fauna in the area directly near the outlet of the pipes, the impact of sedimentary pollutants on the surface of the ocean offshore was

not even mentioned. This aspect is highly worrying since the type of pollutants emitted by the chimneys have direct harmful effects on the composition of the water and on marine life. Performing a search on the Google Scholar bibliographic platform with the concepts "acidification—ocean—Chile", in Spanish and English, it is possible to find more than 10,000 articles that mention the seriousness of the impact of pollutants in the ecosystem marine from the last 3 years [50,51].

Equally important is the absence of more in-depth evaluations regarding the impact of atmospheric emissions on unpopulated areas. The current environmental regulations do not require that the analysis be carried out with respect to the type of soil that the contaminants could affect, but rather establishes the maximum concentration that must be recorded according to the type of contaminant. Note that for this branch of the regulations, there are two regulations in force by 2022, the one referring to the maximum levels of sulfur dioxide that governs at the national level [38], and the one regarding sedimentable particulate matter exclusive for the valley of Huasco in the third region of Chile [39]. This is very different from what is regulated on the impact on the health of the population, where there are seven regulations in force throughout the Chilean territory.

The evaluation of nearly 30 years of projects and/or activities related to the implementation and operation of thermoelectric plants, as well as the updating of laws, regulations, and standards for air quality evaluation, allowed the capacity to show a clear lack of synchrony between their executions. Of the 40 studies evaluated (Table 3), 33 were approved before 2010, in addition to the fact that Supreme Decree 61 was only enacted in 2008, approving the regulation of atmospheric pollutant measurement stations [41], which shows that for at least 20 years, approvals were carried out without a single regulation or methodology aimed at setting the criteria for the use and recording of the information obtained by the monitoring stations.

The creation of government institutions and regulatory entities strengthens the fact that for Chile the emission of pollutants generated by the burning of fossil fuels is a critical problem that needs to be addressed [15,43,45]. The bibliography regarding the negative consequences that this fact has brought to Chile is extensive. However, and even though years have passed since the need to regularize the control processes and generation of atmospheric emissions was made known, and despite the appearance of these entities, the quality of the environmental management instruments (EIS and EIA), as well as their evaluation processes and methodologies, continue to be causes for environmental concern [52–54].

The research exercise carried out in this study made it possible to demonstrate that there is no regulation that establishes as a requirement the installation of new monitoring stations in the areas where the greatest flow of atmospheric pollutants will actually occur, but rather that these will be monitored by air quality stations close to the location where the thermal power plants are installed, which ideally meet the criterion of population representativeness. Accordingly, the measurements made in the inspections of the operation and generation emissions of the plants will not be recording the real concentrations of pollutants expelled into the atmosphere, but only those that are displaced by wind currents to the areas where the air quality monitoring stations are located.

From the point of view of collecting information, it is important to carry out further studies in which more information can be obtained regarding the internal procedures for estimating and projecting atmospheric emissions, which are currently carried out by private consultants whose information is reserved as it is considered industrially sensitive. It is recommended that the collection of information be carried out in situ in the localities where both the emission sources and the monitoring stations are located, since it was only in 2010 that this information began to be required in digital form, so all the previous information may not be available completely digitized on public access platforms.

At the regulatory level, the current air quality regulations need to be updated to include more environmental aspects in the maximum allowable concentration limits. Similarly, the type of surface in which the contaminants would settle at distances greater than the

surrounding populated areas, directly impacting the generation of new secondary quality standards, deserves consideration. The configuration of a single national standard for the regulation of atmospheric emissions is highly necessary in order to unify the criteria of dangerousness of the effect of atmospheric pollutants on the environment.

Finally, there is the need to study the dispersion of atmospheric pollutants generated by thermoelectric power plants, which allows determining the current areas of maximum concentration for the installation of new air quality monitoring stations, whose records allow projecting scenarios with greater representativeness of the atmospheric, environmental, and population situation, given the possibility of new power generation projects.

## 5. Conclusions

This study shows the relevance of the concept of representativeness of monitoring stations and air quality, as well as the methodology of use and interpretation of the data recorded by them. The correct selection of the locations turns out to be crucial for the evaluation and projection of the environmental and population impacts that could be generated as a result of the operation of the thermoelectric power plants. The Chilean national situation reflects that for decades this information has not been obtained with the corresponding rigor, calling into question the results and approvals of projects and/or activities supervised by the legislative bodies.

New studies will be necessary to model with real and reliable data the true situation of the dispersion of atmospheric pollutants in areas categorized as environmental sacrifice zones, with a view to optimizing industrial operations for the benefit of the environment.

**Author Contributions:** Conceptualization, M.C.; Data curation, M.C.; Formal analysis, M.C.; Funding acquisition, G.A.R.; Investigation, M.C. and G.A.R.; Methodology, M.C. and G.A.R.; Project administration, M.C. and G.A.R.; Resources, G.A.R.; Software, M.C.; Supervision, G.A.R.; Validation, M.C. and G.A.R.; Writing—original draft, M.C.; Writing—review and editing, M.C. and G.A.R. All authors have read and agreed to the published version of the manuscript.

**Funding:** This research was partially funded by FONDECYT 1180706, PIA/BASAL FB0002, and TDP220017 grants from ANID, Chile.

**Institutional Review Board Statement:** Not applicable.

**Informed Consent Statement:** Not applicable.

**Data Availability Statement:** The data that support the findings of this study are available from the first (M.C.) and corresponding (G.A.R.) authors, upon request.

**Conflicts of Interest:** The authors declare no conflict of interest.

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
