# Peer review of "Evaluation of Atmospheric Environmental Regulations: The Case of Thermoelectric Power Plants"

_atmosphere, doi:10.3390/atmos14020358_

Round 1

Reviewer 1 Report

atmosphere-2149645

Title:  Evaluation of atmospheric environmental regulations: The case of thermoelectric power plants.

Authors:  Mateo Concha , Gonzalo A. Ruz

This study evaluates efficiency of environmental assessment instruments of the thermoelectric power plants in a sacrifice zone in Chile in regards to their consistency with current environmental quality regulations. The authors consequently define improvement factors for air quality regulations and assessment, and suggest alternative methodologies and criteria for atmospheric regulation with emphasis on the population and environmental health for the sacrifice zones.  The importance of the study stems from highlighting the shortcomings of the current air quality regulations in Chile and the low representativeness of records of air quality monitoring stations. 

The paper reads easy, is understandable and scientifically sound, however, in my opinion this study is out of the scope of the Atmosphere journal.  “Atmosphere is an open access, international, interdisciplinary scholarly journal focused on all areas of scientific research related to atmosphere”; here authors discuss more policy and regulations related to the pollution.

Minor suggestions: 

Lines 34-37:  instead of talking about the process of the dispersion of pollutants, the authors could explain here briefly the composition and toxic effects of pollutants found typically in the emissions from energy industries (in other words why power plant emissions are bad to health and the environment);

Lines 234-235:  “1 hour   1.000”; do you mean 1,000?;

Line 381:  “… distribution of thermal stations and power plants based on environmental conservation.” Did you mean … distribution of monitoring stations…. ?

Reviewer 2 Report

I carefully read the manuscript and I found that the presented results are interesting. However, new advances in the field are missing, including the long-memory effect in air pollution, and the corrosive impacts of air pollutants to historic and modern materials, notably:

1)      In the Introduction the following must be mentioned:

i)                   Atmospheric pollution is an accelerating factor in the material deterioration of buildings and other structures as well as objects of cultural heritage.  Of crucial importance is the corrosive effects of gaseous SO2, NOx, O3, HNO3, particulate matter, and acid rainfall in combination with climatic parameters (see for example, https://doi.org/10.1007/s11356-009-0114-8, https://doi.org/10.5194/acp-11-12039-2011

ii)                 High ultra-fine particle concentrations at the urban site generally coincided with periods of high NO concentrations and were well correlated with benzene, signifying emissions from motor vehicles and industrial regions (see for example, https://doi.org/10.1016/j.atmosenv.2012.05.015

iii)               The atmospheric phenomena such as precipitation, El-Nino, land and sea temperature etc. obey long memory effect, which is employed in their forecasting. In this context, the extensive photochemistry enhancement observed in air-polluted areas for long time-periods exhibit long-memory. It seems that the strength of this memory stems from its temporal evolution and provides the limits of the air pollution predictability at various time scales.

2)      Please add a very brief argument on what is the contribution of your paper to the success of the Sustainable Development Goals-UN 2030 agenda.

In summary, performing the afore-mentioned improvements I do believe that the paper can be considered for publication. 

I will be happy to read the revised version.

Round 2

Reviewer 2 Report

I carefully read the new version of the paper and found that the authors satisfactorily followed all my suggestions.

After that I believe that the current version presents convincing and interesting results which will be attractive to the readers of the journal.

Consequently, I propose to publish the paper in its new form as it is.